# Eco-Friendly 1,3-Dipolar Cycloaddition Reactions on Graphene Quantum Dots in Natural Deep Eutectic Solvent

**DOI:** 10.3390/nano10122549

**Published:** 2020-12-18

**Authors:** Salvatore V. Giofrè, Matteo Tiecco, Consuelo Celesti, Salvatore Patanè, Claudia Triolo, Antonino Gulino, Luca Spitaleri, Silvia Scalese, Mario Scuderi, Daniela Iannazzo

**Affiliations:** 1Department of Chemical, Biological, Pharmaceutical and Environmental Sciences, University of Messina, I-98168 Messina, Italy; sgiofre@unime.it; 2Department of Chemistry, Biology and Biotechnology, University of Perugia, I-06123 Perugia, Italy; matteotiecco@gmail.com; 3Department of Engineering, University of Messina, I-98166 Messina, Italy; ccelesti@unime.it; 4Department of Mathematical and Computer Sciences, Physical Sciences and Earth Sciences, University of Messina, I-98166 Messina, Italy; patanes@unime.it; 5Department of Civil, Energy, Environmental and Materials Engineering, Mediterranea University, I-89122 Reggio Calabria, Italy; claudia.triolo@unirc.it; 6Department of Chemical Sciences, University of Catania and I.N.S.T.M. UdR of Catania, I-95125 Catania, Italy; agulino@unict.it (A.G.); luca.spitaleri@phd.unict.it (L.S.); 7Institute for Microelectronics and Microsystems, National Research Council (CNR-IMM), Ottava Strada n.5, I-95121 Catania, Italy; silvia.scalese@imm.cnr.it (S.S.); mario.scuderi@imm.cnr.it (M.S.)

**Keywords:** graphene quantum dots, 1,3-dipolar cycloadditions, natural deep eutectic solvents, eco-friendly reactions

## Abstract

Due to their outstanding physicochemical properties, the next generation of the graphene family—graphene quantum dots (GQDs)—are at the cutting edge of nanotechnology development. GQDs generally possess many hydrophilic functionalities which allow their dispersibility in water but, on the other hand, could interfere with reactions that are mainly performed in organic solvents, as for cycloaddition reactions. We investigated the 1,3-dipolar cycloaddition (1,3-DCA) reactions of the C-ethoxycarbonyl *N*-methyl nitrone **1a** and the newly synthesized *C*-diethoxyphosphorylpropilidene *N*-benzyl nitrone **1b** with the surface of GQDs, affording the isoxazolidine cycloadducts *isox*-GQDs **2a** and *isox*-GQDs **2b**. Reactions were performed in mild and eco-friendly conditions, through the use of a natural deep eutectic solvent (NADES), free of chloride or any metal ions in its composition, and formed by the zwitterionic trimethylglycine as the -bond acceptor, and glycolic acid as the hydrogen-bond donor. The results reported in this study have for the first time proved the possibility of performing cycloaddition reactions directly to the p-cloud of the GQDs surface. The use of DES for the cycloaddition reactions on GQDs, other than to improve the solubility of reactants, has been shown to bring additional advantages because of the great affinity of these green solvents with aromatic systems.

## 1. Introduction

Graphene quantum dots (GQDs), zero-dimensional carbon-based nanomaterials with graphene structures, have received significant interest from academia and industry in recent years [1]. Their outstanding physicochemical properties enabled novel and extraordinary applications in several fields including physics, chemistry, materials science, biology and medicine [2]. Unlike two-dimensional graphene, GQDs show a band-gap, because of the quantum size effect, as well as stable and size-dependent photoluminescence. Both band-gap and photoluminescence can be tuned during the GQDs synthesis by controlling their size, shape, charge transfer between functional groups and graphene surface, and by doping with heteroatoms [3]. Because of their unique optical properties, these nanomaterials have also been designed for application in photovoltaics, for the development of flexible devices and in biosensing [4,5]. Moreover, their low toxicity, high water dispersibility, large surface to volume ratio and versatile surface functionalization with several functional groups and biologically active molecules make these nanoparticles ideal nano-carriers for drug delivery and bioimaging [6,7]. GQDs have been functionalized with drugs and labeled with tumor-targeting ligands able to specifically bind cancer receptors exposed on the cancer cells surface, thereby developing new therapies for a more effective delivery of toxic anticancer drugs while minimizing their impact on healthy tissues. Moreover, their photoluminescence properties allowed the development of bioimaging agents with high sensitivity in both in vitro and in vivo models of different types of cancer and advanced tools for theranostics applications [6,7,8].

The chemical functionalization of GQDs plays a key role for the development of these nanomaterials and, in recent years, a great number of studies have reported their modification at the graphitic skeletons or at the edges, doping with heteroatoms, hybridization with metallic nanostructures and assembly with other materials, thus modifying the GQDs structure and properties for the desired application [9]. Analogously to the other members of the graphene family, the presence of multiple sites for GQDs functionalization, allows the covalent binding with organic functionalities or biomolecules to the graphene surface, or to oxygen groups available after oxidative treatments. In addition, non-covalent electrostatic interactions, such as the π−π interaction between aromatic compounds and the graphene structure, can be pursued [10,11,12]. The various top-down and bottom-up methodologies for the synthesis of GQDs have been shown to regulate the physical, chemical and biological properties of these nanomaterials [6]. In particular, GQDs synthesized by top-down strategies generally possess many hydrophilic functionalities which allow their dispersibility in water and make these nanomaterials less biologically reactive and easier to be transported along the biological milieu, thus favoring their use for biomedical applications [6]. However, the presence of hydrophilic functionalities can interfere with common synthetic reactions that are mainly performed in organic solvents. In particular, cycloaddition reactions normally used for the direct functionalization of graphene-based materials, such as graphene [13,14,15] carbon nanotubes [16,17,18] and fullerene [19,20], can hardly be performed in water, because of the general low solubility of organic reactants in water-based solvents [21] and, as a consequence, for GQDs, the peripheral edge-functionalization is mainly reported [22,23]. 

Among the different water-soluble liquid systems, deep eutectic solvents (DESs), formed from a eutectic mixture of mainly solid hydrogen bond donor and hydrogen bond acceptor compounds, represent an environmentally friendly alternative to hazardous organic solvents. These green solvents are biodegradable, reusable and do not require purification procedures for their synthesis [24]. These innovative H-bond based liquids have been used in biocatalyzed, and organocatalyzed reactions and also as solvents for 1,3-dipolar cycloadditions [25,26,27], thereby proving to overcome problems related to the use of water insoluble dipoles and improving the reaction selectivity in metal-free conditions [28]. Thanks to their polarity, DESs have also shown an interesting behavior in microwave assisted reactions [29]. Noteworthy, the use of DESs for cycloaddition reactions on GQDs meets additional advantages because of the great affinity of these solvents with aromatic compounds [30].

Here, we report for the first time the microwave-assisted 1,3-dipolar cycloaddition (1,3-DCA) of two nitrones: *C*-ethoxycarbonyl *N*-methyl nitrone **1a** and the newly synthesized *C*-diethoxyphosphorylpropilidene *N*-benzyl nitrone **1b**—with the graphene surface of GQDs affording eco-friendly and mild reaction conditions, the isoxazolidine cycloadducts *isox*-GQDs **2a** and *isox*-GQDs **2b** (Figure 1). Nitrone **1a** was chosen for its known high reactivity towards different dipolarophiles [31], in order to test its reactivity with nano-scaled dipolarophiles and for the presence of the further functionalizable carboxyethyl group. Nitrone **1b** was also investigated for synthetic purposes, as it possess a phosphonate group, suitable for the further anchorage with nucleic acids, for gene delivery applications. 

The reactions were carried out using a DES obtained from a natural source (natural deep eutectic solvent—NADES) [32], formed by the zwitterionic trimethylglicine (TMG) as hydrogen-bond acceptor (HBA) and the glycolic acid (GA) hydrogen-bond donor (HBD). The used NADES does not contain chloride or metal ions in its composition, with both components derived from sugar beet, and is cheap, recyclable and shows advantageous physical properties such as low viscosity and low melting point [32]. The use of this green solvent has been shown to exert a positive effect for the reactions’ outcomes, also improving the solubility of the water insoluble reactants and of GQDs, because of the great affinity of the solvent with the carbon sp^2^ network. The synthetic strategy allowed the introduction of a carboxyethyl or a phosphonic group as new sites for the further functionalization of GQDs and, of course, can be extended to other dipoles for the introduction of various substituents at R_1_ and R_2_ positions.

## 2. Materials and Methods 

### 2.1. Materials

Solvents and chemical reagents were obtained from commercial suppliers and used as received, without any further purification. The multi-walled carbon nanotubes (MWCNTs) used for the synthesis of GQDs were produced as previously reported [33], from isobutene by catalytic chemical vapor deposition, using Fe/Al_2_O_3_ as the catalyst. GQDs were synthesized by acidic treatment from MWCNTs, using a mixture of HNO_3_/H_2_SO_4_ (1:3 ratio), following a reported procedure [34].

### 2.2. Chemical, Physical, and Morphological Characterization

^1^H NMR spectra were registered using a 500 MHz Varian instrument; proton chemical shifts have been reported in ppm (δ), from tetramethylsilane (TMS) as the internal standard. Microwave reactions were carried out with a Discover Focused Microwave System (CEM Corporation, NC, USA). Thin-layer chromatography was performed on Merck silica gel 60-F254 precoated aluminum plates while preparative separations were carried out through flash chromatography using Merck silica gel of 0.063–0.200 mm and 0.035–0.070 mm. Micro Raman measurements were carried out using a confocal microscope NT-MDT NTEGRA Spectra (NT-MDT Spectrum Instruments, Moscow, Russia) in reflection mode, exciting the sample with a Nd:YAG laser at the λ_exc_ of 532 nm. The GQDs morphology was evaluated by transmission electron microscopy (TEM). To make the specimen suitable for TEM observation, an aqueous solution containing GQDs was dropped out on a lacey-carbon TEM grid. TEM analysis was performed in a probe aberration-corrected JEOL JEM-ARM200CF microscope (JEOL USA, Inc., Peabody, MA, USA), operated at a primary beam energy of 60 keV. Size characterizations of the synthesized samples was performed by dynamic light-scattering (DLS) analysis using the instrument Zetasizer 3000 (Malvern Panalytical Ltd, Worcestershire, UK), equipped with a 632 nm HeNe laser, operating at a 173° detector angle. UV spectra have been performed with a Thermo Nicolet mod, Evolution 500 spectrophotometer (Thermo Fisher Scientific, Waltham, MA, USA). Photoluminescence (PL) measurements were performed using a NanoLog modular spectrofluorometer Horiba (Horiba Scientific, Kyoto, Japan), at room temperature and with a xenon lamp as excitation source; the GQD based nanomaterials were used at the concentration of 100 ng/mL. Thermogravimetric analyses were performed in argon atmosphere at 10 °C/min, from 100 to 1000 °C, using a TA Q500 instrument (TA Instruments, New Castle, DE, USA). The infrared spectra were registered using a Perkin Elmer Spectrum 100 spectrometer (Perkin Elmer Italia S.p.A., Milano, Italy), equipped with a universal ATR sampling accessory; the spectra were recorded at room temperature and without any preliminary treatment, from 4000 to 600 cm^−1^, with a resolution of 4.0 cm^−1^. X-ray photoelectron spectra (XPS) were performed on a PHI 5600 Multi Technique System (RBD Instruments Inc., Oregon, USA), at an incident angle of 45° relative to the surface plane and a base pressure of the main chamber of 1 × 10^−8^ Pa [35,36]. The samples were excited using Al Kα X-ray radiation with a pass energy of 5.85 eV. The energy resolution of the instrument was ≤0.5 eV. The structures due to the Kα satellite radiations have been subtracted from the spectra before the data processing. XPS peak intensities were calculated after removal of Shirley background [35,36]. The analysis of atomic concentrations was carried out by considering the relative sensitivity factors. The calibration of spectra was performed by setting the C 1s signal at 285.0 eV [35,36]. Some XP spectra were fitted with a symmetric Gaussian line-shape, after subtraction of the background. The process requires the refinement of data, based on the least squares fitting technique, accomplished until the highest possible correlation between the theoretical profile and the experimental spectrum was observed. The value of the agreement or residual factor R, defined by R= [Σ(F_obs_ − F_calc_)^2^/ Σ (F_obs_)^2^]^1/2^, after minimization of function Σ(F_obs_ − F_calc_)^2^, converged to ~0.03. Samples for XPS measurements were prepared by deposition and drying of a few drops of their aqueous solutions on silicon substrates. Atomic force microscopy (AFM) images of the samples surface were obtained with a microscope NT-MDT NTEGRA Spectra and using a Si-cantilever operating in semi-contact mode.

### 2.3. Synthesis of GQDs

The starting GQDs were synthesized by prolonged acidic oxidation and exfoliation of pristine MWCNTs using a solution of HNO_3_/H_2_SO_4_ in 1:3 ratio. The mixture, placed in a reaction vessel equipped with a water condenser, was sonicated, at 60 °C for 4 days. Then, after dilution with deionized water, the suspension was filtered under vacuum through a 0.1 µm Millipore membrane. A NaOH solution was added to the filtrate until neutral pH and, after dilution with water, the suspension was transferred in a dialysis bag (12,000 Dalton molecular weight) for the sample purification. A little amount of the resulting suspension was dried at 60 °C, under vacuum, and used for the further characterizations. The number of acidic groups present on the GQDs surface was found to be of 2.37 mmol/g, as calculated by titration analysis using the Zetasizer 3000 instrument. 

### 2.4. Synthesis of NADES

The natural deep eutectic solvent was prepared following a procedure reported in previous papers [37]. The solid components were weighted in a flask at the proper molar ratio (2/1 glycolic acid/trimethylglicine) and then mixed and heated at 50 °C until a homogeneous liquid was formed (generally after 30 mins).

### 2.5. Synthesis of C-(diethoxyphosphoryl)propylidene, N-benzyl nitrone 1b

Triethyl phosphite (5 mL, 28.9 mmol) was added dropwise to an excess of neat 2-(2-bromoethyl)-1,3-dioxolane 3 (6.5 mL, 57.8 mmol) at room temperature and under argon. Then, the mixture was heated up to 110 °C until the NMR analysis of the reaction mixture confirmed the complete disappearance of triethyl phosphite. The diethyl ethylphosphonate derivative pale oil 4 was obtained by distillation, through a Vigreux column under reduced pressure, of the crude reaction mixture (yield 58%). ^1^HNMR (CDCl_3_) *δ* = 4.95 (t, *J* = 4 Hz, PCH_2_CH_2_C*H*O_2_, 1H), 4.06–4.15 (m, OC*H_2_*CH_3_, 4H), 3.84–3.99 (m, OC*H_2_*C*H_2_*O, 4H), 1.79–2.00 (m, PC*H_2_*C*H_2_,* 4H), 1.32 (t, *J* = 7 Hz, OCH_2_C*H_3,_* 6H); ^13^C NMR (75.5 MHz; CDCl_3_) *δ* = 103.7 (d, ^3^*J*_P–C–C–C_ = 19), 65.5, 61.9 (d, ^2^*J*_P–O–C_ = 6 Hz), 27.3 (d, ^2^*J*_P–C–C_ = 4 Hz), 19.9 (d, ^1^*J*_P–C =_ 144 Hz), 16.8 (d, ^3^*J*_P–O–C–C_ = 6 Hz). An aqueous HCl 2 M and acetone 10:1 v:v solution (50 mL) of diethyl [2-(1,3-dioxolan-2-yl)ethyl]phosphonate 4 (3 g, 12.6 mmol) was heated for 3 h at 50 °C. Then, after cooling, the reaction mixture was extracted using CH_2_Cl_2_ (3 × 50 mL); the organic layers were dried over MgSO_4_ and concentrated under vacuum to give the aldehyde 5 as a pale oil (88% yield) that was used for the successive reaction without any further purification. To a solution of sodium acetate (1,2 g, 15 mmol) in CH_2_Cl_2_ (30 mL), cooled at 0 °C, were added the *N*-benzylhydroxylamine hydrochloride (2.4 g, 15 mmol) and successively the aldehyde 5 (2 g, 10 mmol) dropwise. The reaction mixture was then stirred for 1 h at 0 °C and then at room temperature overnight. After this time, the solvent was removed under reduced pressure and the residue was purified by silica gel flash chromatography (CHCl_3_/MeOH 95:5) to give the pure nitrone 1b (yield 95%). ^1^H NMR (500 MHz, CDCl_3_) δ = 7.47–7.11 (m, *H*-benzen + PCH_2_CH_2_C*H,* 6H), 4.17–3.91 (m, Ph*CH_2_*+ OC*H_2_*CH_3_, 6H), 3.11–2.91 (m, PCH_2_C*H_2_*CH*,* 2H), 1.98–1.74 (m, PC*H_2_*CH_2_, 2H), 1.36–1.19 (m, OCH_2_C*H_3,_* 6H). ^13^C NMR (125 MHz, CDCl_3_) δ = 139.5 (d, ^3^*J*_P–C–C–C_ = 9 Hz), 137.3, 129.8, 128.2, 128.1, 69.0, 61.6 (d, ^2^*J*_P–O–C_ = 6 Hz), 34.5 (d, ^2^*J*_P–C–C_ = 7 Hz), 26.8 (d, ^1^*J*_P–C_ = 94.9 Hz), 16.3 (d, ^3^*J*_P–O–C–C_ = 6 Hz).

### 2.6. Synthesis of isox-GQDs **2a** and isox-GQDs **2b**

A water solution of GQDs (30 mg/30mL) was treated with a mixture of GA/TMG in 2:1 molar ratio (2 mL). Then, after removal of water under reduced pressure, 300 mg of nitrone **1b** or **2b** were added and the mixture was heated under microwave irradiation for 1 h at 90 °C, 150 W. The obtained suspension was diluted with deionized water and then purified through dialysis for 2 days, using a dialysis bag (MW of 12,000 Da). The degree of functionalization was evaluated on a known amount of sample by TGA under argon atmosphere for 2 days. The quantity of the unreacted nitrones was also evaluated by extraction with ethyl acetate (3 × 3 mL) from the dialysis water solutions containing NADES. The organic phase was dried over anhydrous magnesium sulphate and the solvent was evaporated under reduced pressure. The presence of nitrones **2a** or **2b** was confirmed by ^1^H NMR characterization.

## 3. Results and Discussion

The GQDs used for this study were synthesized by a top-down procedure previously reported by us [34], starting from multi-walled carbon nanotubes (MWCNTs) in order to obtain nanodots with many oxygen-containing functional groups [38] and were characterized by Raman, high-resolution transmission electron microscopy (HRTEM), dynamic light scattering (DLS), UV-VIS, and photoluminescence (PL) analyses. Raman spectra (Appendix A) show the D-band (ca. 1320 cm^−1^) and G-band (ca. 1590 cm^−1^), usually found in carbon nanostructures. Their relative intensity ratio (I_D_/I_G_ ratio) showed values of 1.03 for the starting MWCNTs and of 1.68 for the GQDs, thus demonstrating the loss of long-range order after the top-down procedure. The representative TEM image of GQDs shows monodisperse nanoparticles with a weighted size distribution centered at 4.8 nm; moreover, the HRTEM image in the inset further confirms the lattice fringes (≈ 0.21 nm) of GQDs (Figure 2a). DLS measurements showed a maximum of the volume weighted percent at 5.85 nm, thus further demonstrating the small size of the synthesized nanomaterials (Figure 2b). Figure 2c shows the UV-vis absorption and photoluminescence excitation of the synthesized nanomaterials. The UV spectrum shows the typical π–π transition absorption peak at around 250 nm, due to the π–π* transition of the aromatic sp^2^ domains, while PL measurements confirm the emission properties of GQDs because at the excitation wavelength of 360 nm the nanomaterials water dispersion showed a strong peak at 560 nm.

The direct graphene functionalization of GQDs by 1,3-DCA was exploited using two different nitrones: the previously reported nitrone **1a** [39] and the newly synthesized phosphonated nitrone **1b**. Nitrone **1b** was synthesized starting from 2-bromoethyl-1,3-dioxolane 3 which was subjected to the Michaelis–Arbuzov reaction with triethyl phosphite to give the corresponding phosphonate 4 [40] and then, after deprotection of the cyclic acetal and reaction of the obtained aldehyde **5** with *N*-benzyl hydroxylamine, afforded **1b** (see Appendix A). The 1,3-DCA was carried out under microwave irradiation, at 90 °C for 1 h using an excess of **1a** or **1b** (10: 1 wt% with respect to GQDs) and, as a deep eutectic solvent, a mixture of GA/TMG in 2:1 molar ratio which, on the basis of a previous study [37], represents the optimal eutectic ratio (Scheme 1).

At the end of each experiment, the unreacted nitrone was extracted with ethyl acetate from the washing water solutions after dialysis, in order to evaluate the degree of functionalization and to demonstrate the recycling capabilities of the tested solvent. The recycling of green solvents is in fact an important issue in order to increase the sustainability of a chemical process. As reported in other studies, the water solubility of NADES can be exploited for their easy recovery by simple water extraction and also for their reuse [37,41].

After removal of unreacted reagents and DES by dialysis, the effectiveness of the reactions leading to the formation of the cycloadducts *isox*-GQDs **2a** and *isox*-GQDs **2b**, was investigated by Fourier transform infrared spectroscopy (FTIR), thermogravimetric analysis (TGA) and by X-ray photoelectron spectroscopy (XPS) analyses. The chemical and physical properties of the synthesized nanomaterials were evaluated by Raman spectroscopy, DLS and PL analyses while their morphology was investigated by atomic force microscopy (AFM). The TGA curves of GQDs and of the corresponding cycloadducts, performed under inert atmosphere, show for both cycloadducts, an increase of weight losses, whose amounts, as calculated at 500 °C, were found to be of 8.4 and 4.9 wt % for *isox*-GQDs **2a** and *isox*-GQDs **2b,** respectively (Figure 3a). Moreover, the different profiles of TGA curves of cycloadducts confirm that deep chemical modifications occurred on the nanomaterials after the chemical functionalization. The degree of functionalization was also confirmed by calculating the difference between the initial amount of nitrone used to functionalize GQDs and the amount of unbound dipole present in the washing water solutions, after dialysis. The calculated percentages of 8.3 and 5.0 wt% for *isox*-GQDs **2a** and *isox*-GQDs **2b**, respectively, are in agreement with the results obtained from TGA analyses.

The FTIR spectrum of GQDs shows the presence of a large band at around 3450 cm^−1^, related to the stretching of O−H bonds, the stretching of the C=O group at 1610 cm^−1^ and the bending of the O−H group at 1420 cm^−1^. These peaks indicate the presence of many oxygen-containing groups on the GQDs surface. The FTIR spectra of the cycloadducts show the additional representative peaks at 850 cm^−1^ related to the stretching of the newly formed N−O bond. Moreover, the spectrum of *isox*-GQDs **2a** shows the presence of a peak at 1727 cm^−1^, related to the newly introduced ester functionality, while for the phosphonated adduct *isox*-GQDs **2b**, the additional peak at 958 cm^−1^, ascribable to the vibration of the P−O group, can be observed (Figure 3b). 

The electronic structure of the pristine and functionalized GQDs was investigated through XPS, which provides information on the different functional groups and allows quantitative estimation, once the relevant atomic sensitivity factors are taken into account [35]. As an initial check control, we performed XPS analysis of GQDs (see Appendix A). Appendix A show their high-resolution XP spectra in the C 1s and O 1s binding energy (B.E.) regions. A careful fitting of the experimental profile of the C 1s signal required three Gaussian components centered at 285.0, 286.7, and 288.7 eV, respectively (Appendix A). The first component (285.0 eV) is due to the aromatic and aliphatic backbones [35]. The other peaks at 286.7 and 288.7 eV are assigned to the C–OH and –COOH functional group, respectively [35]. Worthy of note, the intensity ratio between these last two bands (4:3) nicely fits the –C–OH and –COOH functionalities in the GQDs material. The O 1s spectral profile was fitted using four Gaussian components at 531.5, 532.7, 533.7, and 535.4 eV, respectively (Appendix A). The lower energy peak, detected at 531.5 eV is due to the oxygen of the C=O groups of the GQDs [35]. The second peak at 532.7 eV is assigned to –OH groups and oxygen of SiO_2_ substrate, whereas the peak located at 533.7 eV is assigned to the OH oxygen of the carboxylate groups [35]. The higher energy peak located at 535.4 eV is attributed to some H_2_O molecules present on the GQDs [35]. Once more, the intensity ratios of the first three peaks are in agreement with the expected intensity trend (1:2:1) on the basis of the GQDs composition. Figure 4a shows the high-resolution XP spectrum of *isox*-GQDs **2a** in the C 1s energy region. The accurate fitting of the spectrum profile showed the presence of three components at 285.0, 286.5 and 288.5 eV. The first component (285.0 eV) is due to both aromatic and aliphatic backbones. The second peak at 286.5 eV is due to the C–N and C–OH groups [37]. Finally, the peak at 288.5 eV is assigned to the –COOH groups [35]. Figure 4b shows the O1s spectrum for *isox*-GQDs **2a**, fitted using four Gaussian components at 531.7, 532.8, 533.9, and 535.1 eV, respectively [35]. These XPS values are almost coincident with those observed for the GQDs sample. Figure 4c shows the XP spectra of the GQDs functionalized with **1a** or **1b** in the N 1s binding energy region. The XPS of both cycloadducts show only one clear band at 400.3 eV, attributed to the nitrogen of the isoxazolidine ring, which confirms the functionalization of the GQDs with 1b [42,43,44]. Figure 4d shows the high-resolution XP spectrum of *isox*-GQDs **2b**, in the C 1s energy regions. An accurate fitting of the spectrum revealed the presence of three components at 285.0, 286.7, and 288.8, respectively. These values are almost coincident with those observed for the GQDs blank sample. In fact, the first component (285.0 eV) is due to both aliphatic and aromatic backbones [35]. The second peak at 286.7 eV is now due to both C–N and C–OH groups of nitrone GQDs and the peak at 288.8 eV is assigned to the –COOH groups of GQDs [35]. The increased intensity of the XPS band at 286.7 eV, with respect to that observed in the blank sample, confirms the functionalization of GQDs with the phosphonate nitrone. In particular, the intensity ratio between the two bands at 286.7/288.8 eV is now 6:3 whilst in the blank sample was 4:3. This increase is consistent with two –C-N moieties of a nitrone unit. Figure 4e shows the O1s spectrum for *isox*-GQDs **2b**, fitted using four Gaussian components at 531.6, 532.7, 534.0, and 535.3 eV, respectively [35]. These B.E. values are in agreement with the low degree of functionalization (4.7 wt %) observed by TGA analysis. Figure 4f shows the XPS of the P 2p states for the *isox*-GQDs **2b**. The noisy peak at 133.4 eV accounts for the ionization of the –PO_3_C_2_H_5_ [45]. As far as the intensity of this band is taken into account, it emerges that the phosphonate group in GQDs is very low as also observed by TGA analysis. The XPS atomic concentration analyses performed on the two functionalized samples gave a N 1s **2a**/**2b** ratio of 2.4, thus confirming the larger functionalization of *isox*-GQDs **2a**. 

We have evaluated the effect of the surface modification on the dimensions and the water dispersibility of the cycloadducts, which deeply affect the interaction of nanomaterials with biological system [46] (Appendix A). Both samples *isox*-GQDs **2a** and *isox*-GQDs **2b** show single size populations centered at 6.61 and 4.95 nm, respectively. The water dispersibility, evaluated by measuring their electrophoretic mobility in deionized water, showed zeta potential values lower than −30 mV (−30.7 mV and −31.2 mV for *isox*-GQDs **2a** and *isox*-GQDs **2b**, respectively) thus further confirming their high stability in water [47]. 

The surface modification of GQDs was also investigated by comparing the PL properties of the starting nanomaterials with those of the corresponding cycloadducts in deionized water at the excitation wavelength of 360 nm (Figure 5). The PL λ_max_ of GQDs, *isox*-GQDs **2a** and *isox*-GQDs **2b** were about 560, 545, and 550 nm, respectively. These blue-shifted emissions can prove that organic moieties are covalently bound with GQDs, as also reported for similar systems [48].

The study of the Raman spectrum is very useful for understanding the disordered graphitic materials [49]. Figure 6 shows the best fit of the Raman spectra of the GQDs and of *isox*-GQDs **2a** and *isox*-GQDs **2b**. The D- and G- bands dominate all spectra. The D-band is due to the breathing mode of aromatic rings in the carbon network and is related to the phonons of A_1g_ symmetry at the K point of the Brillouin zone and it is disorder-activated [50,51]. The G-band is considered as the Raman fingerprint of the graphitic crystalline arrangements, and it is originated from the E_2g_ symmetry stretching of all sp^2^ bonded C=C pairs [50,51]. Since the D-band progressively intensifies (relative to the G-band) with the increasing deviation from the perfect hexagonally-organized planar carbon network, the I_D_/I_G_ ratio is commonly used to evaluate the presence of disorder in sp^2^ hybridized carbon systems [52,53]. The fitting procedure of the Raman spectra underlines the structural variations of samples after the functionalization process. Moreover, the presence of weak and broad bands centred approximately at 1150−1230 cm^–1^ (T-band) and 1450−1530 cm^–1^ (A-band) is highlighted. The former is attributed to the presence of trans-poly-acetylene-like chains, which are formed at the zigzag edges of defective graphitic layers [50,52,53]. The A-band indicates amorphous phases, connected to the aromatic-rings planes, through Csp^3^ bonds [50,52,53]. The functionalization of the GQDs sample seems to enhance the crystalline quality with respect to the untreated-GQDs, as demonstrated by the narrowing of the D-band after the functionalization process. The GQDs D-band exhibits an FWHM of 176.77 cm^–1^ with a Gaussian shape. After the functionalization, the D-band needs to be fitted by a Lorentzian function and its FWHM is reduced down to 120.47 cm^–1^ for *isox*-GQDs **2a** and 91.37 cm^–1^ for *isox*-GQDs **2b**. In addition, the I_D_/I_G_ ratio progressively decreases from 1.15 for GQDs to 1.06 for *isox*-GQDs **2a** and 0.97 for *isox*-GQDs **2b**, thus indicating a reduction of the carbon defects density, supported by the functionalization process of the GQDs. The ratio of the relative intensities (I_T_+I_A_)/I_D_ is a useful parameter to monitor the “defects-bands” [52], in particular the non-sp^2^ to sp^2^ density ratio. This ratio is lower for the *isox*-GQDs **2a** (its value is 0.26), while is almost constant for GQDs and *isox*-GQDs **2b** (0.44 and 0.42, respectively).

Figure 7 shows the AFM images of GQDs, *isox*-GQDs **2a** and *isox*-GQDs **2b** samples (Figure 7A,C,E) and their line profile (Figure 7B,D,F). All samples consist of few layers of graphene, since the height of the GQDs is about 2–3 nm, independently to the presence of the functional groups.

## 4. Conclusions

The results reported in this study have proved for the first time, the possibility to perform 1,3-DCA on the p-cloud of the graphene surface of GQDs, in mild and eco-friendly reaction conditions, through the use of a deep eutectic solvent, free of chloride or any metal ions in its composition. The synthesized nanomaterials have been shown to maintain the chemical, physical and morphological properties of the starting nanomaterials. Moreover, the introduction of an ester or phosphonate moiety could open new ways to the use of these nanomaterials for gene and drug delivery as well as for imaging or theranostics purposes. The reported synthetic strategy, which allows the introduction of new sites for further functionalization procedures, can be extended to other nitrones, starting from differently substituted hydroxylamines and carbonyl compounds, for the introduction of various alkyl or aryl substituents at R_1_ and R_2_ positions, as well as to other dipoles, using different mixtures of these green deep eutectic solvents.

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
