# Peer review of "Eco-Friendly 1,3-Dipolar Cycloaddition Reactions on Graphene Quantum Dots in Natural Deep Eutectic Solvent"

_nanomaterials, 2020, doi:10.3390/nano10122549_

Round 1

Reviewer 1 Report

According to my opinion the manuscript entitled "Eco-Friendly 1,3-Dipolar Cycloaddition Reactions on Graphene Quantum Dots in Natural Deep Eutectic Solvent", given by Salvatore V. Giofrč et al.  merits to be published by Nanomaterials.  

However, it is not due to using new and eco-friedly solvent, but because of organic synthesis, cycloaddition and new possibilities as far as functionalization of NGO is concerned. Due to the above mentioned facts, this manuscript is really interesting and innovative.

Moreover, modified NGO is characterized very well.

On the contrary, I’m not convinced about the solvent. Is recycling of this solvent possible? According to me, it’s impossible. In such a case, the adjective, namely "eco-friendly" could be an exaggeration. Therefore, the Authors should explain if the solvent recycling is possible and how it can be realized. Adding such an element would be required by me.

Reviewer 2 Report

The work by Iannazzo and coworkers presents a green synthetic method for functionalizing of graphene quantum dots (GQDs) through microwave-assisted 1,3-dipolar cycloaddition in deep eutectic solvents. There are significant amounts of effort on the characterization of the functionalized GQDs. Authors claimed that this is the first [3+2] cycloaddition reaction of p-cloud on the graphene surface of GQDs, which may serve as a linkage reaction for introduce series of functional groups with potential biological and medical utilities.  I recommend the paper for publication after revisions.

  • There are extensive qualitative structure analyses, but experiment and discussion on quantitative analysis (loading or degree of functionalization) of the GQDs are limited.
  • This work established the 1,3-cycloadditon of GQDs, the scope should be expended to including more 1,3-dipolars instead of only two 1a & 1b.
  • Deep eutectic solvents are strong microwave absorbers. Synthesis under MW heating for 1 h at 90 oC is pretty long. Have attempted the reaction at higher temp to reduce time?
  • English and editorial issues:
  1. Many sentences are long and in fragmental form. May consider to reword them. For example, P2, L80 “Among the different water soluble liquid systems, the deep eutectic solvents (DESs), formed from a eutectic mixture of mainly solid hydrogen bond donor and hydrogen bond acceptor compounds, represent an environmentally friendly alternative to hazardous organic solvents, due to their biodegradability, reusability and the absence of purification procedures for their synthesis [23]”
  2. All the compound #s should be in bold
  3. Many places “1,3-dipolar” are written as ““1,3 dipolar”
  4. P1, L43, change “over the last years” to “in recent years”?
  5. P2, L91, should be “the microwave-assisted 1,3-dipolar cycloaddition”
  6. P5, L198, should be “2 g,”
